# *Frailty-adjusted therapy in Transplant Non-Eligible patients with newly diagnosed Multiple Myeloma (FiTNEss (UK-MRA Myeloma XIV Trial)): a study protocol for a randomised phase III trial*

Amy Beth Coulson ![ORCID],[1] Kara-Louise Royle ![ORCID],[1] Charlotte Pawlyn,[2] David A Cairns,[1] Anna Hockaday,[1] Jennifer Bird,[3] Stella Bowcock,[4] Martin Kaiser,[5,6] Ruth de Tute,[7] Neil Rabin,[8] Kevin Boyd,[6] John Jones,[9,10] Christopher Parrish,[11] Hayley Gardner,[12] David Meads,[13] Bryony Dawkins,[13] Catherine Olivier,[1] Rowena Henderson,[1] Phillip Best,[1] Roger Owen,[7] Matthew Jenner,[14] Bhuvan Kishore,[12] Mark Drayson,[15] Graham Jackson,[16] Gordon Cook[1,17]

For numbered affiliations see end of article.

**Correspondence to**
Professor Gordon Cook;
g.cook@leeds.ac.uk

## ABSTRACT

**Introduction** Multiple myeloma is a bone marrow cancer, which predominantly affects older people. The incidence is increasing in an ageing population.

Over the last 10 years, patient outcomes have improved. However, this is less apparent in older, less fit patients, who are ineligible for stem cell transplant. Research is required in this patient group, taking into account frailty and aiming to improve: treatment tolerability, clinical outcomes and quality of life.

**Methods and analysis** Frailty-adjusted therapy in Transplant Non-Eligible patients with newly diagnosed Multiple Myeloma is a national, phase III, multicentre, randomised controlled trial comparing standard (reactive) and frailty-adjusted (adaptive) induction therapy delivery with ixazomib, lenalidomide and dexamethasone (IRD), and to compare maintenance lenalidomide to lenalidomide+ixazomib, in patients with newly diagnosed multiple myeloma not suitable for stem cell transplant. Overall, 740 participants will be registered into the trial to allow 720 and 478 to be randomised at induction and maintenance, respectively.

All participants will receive IRD induction with the dosing strategy randomised (1:1) at trial entry. Patients randomised to the standard, reactive arm will commence at the full dose followed by toxicity dependent reactive modifications. Patients randomised to the adaptive arm will commence at a dose level determined by their International Myeloma Working Group frailty score. Following 12 cycles of induction treatment, participants alive and progression free will undergo a second (double-blind) randomisation on a 1:1 basis to maintenance treatment with lenalidomide+placebo versus lenalidomide+ixazomib until disease progression or intolerance.

**Ethics and dissemination** Ethical approval has been obtained from the North East—Tyne & Wear South Research Ethics Committee (19/NE/0125) and capacity and capability confirmed by local research and development departments for each participating centre prior to opening to recruitment. Participants are required to provide written informed consent prior to trial registration. Trial results will be disseminated by conference presentations and peer-reviewed publications.

**Trial registration number** ISRCTN17973108, NCT03720041.

## STRENGTHS AND LIMITATIONS OF THIS STUDY

⇒ Frailty-adjusted therapy in Transplant Non-Eligible patients with newly diagnosed Multiple Myeloma (FiTNEss) will provide the first prospective data investigating the use of the International Myeloma Working Group frailty score to define appropriate dose delivery strategies for older patients.

⇒ FiTNEss will explore the impact of dual-agent maintenance compared with the single-agent standard of care using a gold standard placebo-controlled design.

⇒ The trial has the potential to meet a high unmet need in older patients with myeloma for whom the impact of recent therapeutic have been less marked.

⇒ Wide inclusion criteria have been designed to maximise the recruitment of older, more frail patients who may not previously have been included in clinical trials.

⇒ Owing to the nature of assessments and dose adaptations, blinding in the induction phase of the trial is infeasible.

## INTRODUCTION
### Multiple myeloma

Multiple myeloma is the second most common haematological malignancy with over 5500 patients diagnosed in the UK each year.[1] Myeloma is predominantly a disease of older people, with two-thirds of patients aged over 70 years at diagnosis. The incidence is increasing as the population ages.

**BMJ**

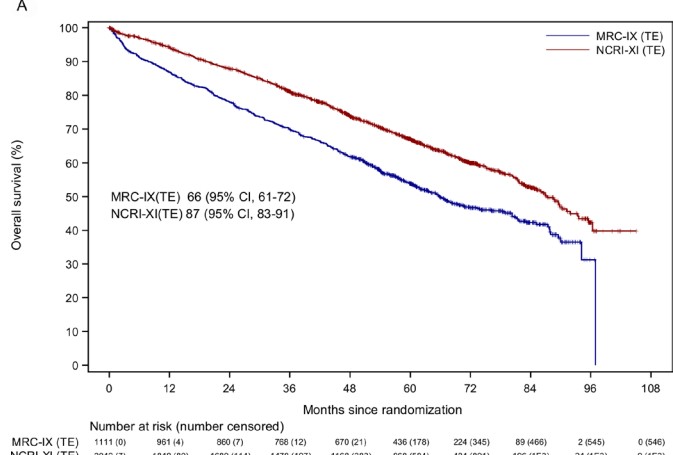

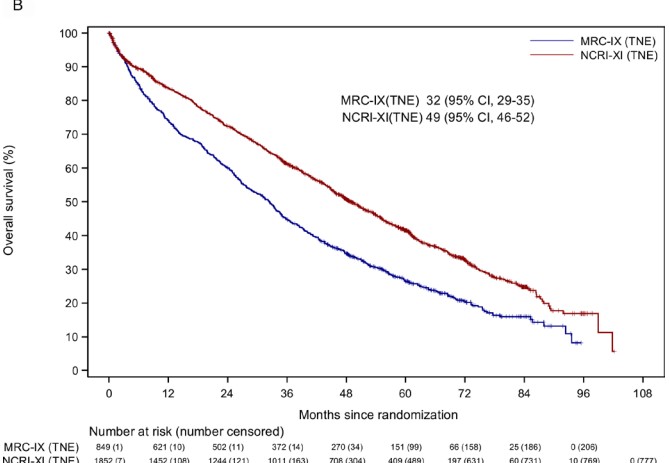

**Figure 1** Overall survival in MRC-IX and NCRI-XI TE pathway (A). Overall survival in MRC-IX and NCRI-XI TNE pathway (B). MRC-IX, Medical Research Council Myeloma IX trial (ISRCTN49407852); NCRI-XI, National Cancer Research Institute Myeloma XI Trial (ISRCTN68454111); TE, transplant eligible; TNE, transplant non-eligible.

Over the last 10 years, the development of proteasome inhibitors (PI), immunomodulatory drugs (IMiD agents) and improved supportive care, have ameliorated outcomes for patients with myeloma such that the median overall survival (OS) is now more than 6 years for younger, fitter patients (figure 1A).[2] However, the impact of these

therapies has been less marked in the older or less fit population, particularly those over 75 years of age and/or ineligible for stem cell transplant (figure 1B). While outcomes in younger patients are largely driven by molecular risk factors present in the myeloma cell clone, there is no evidence that older patients with myeloma have more biologically high-risk disease.[3] Differences in outcomes are likely to be accounted for by changes in patient physiology and/or increased treatment-related toxicity. Data from our previous trial Myeloma XI (ISRCTN49407852) show that as age increases, the number of participants ceasing treatment due to choice or toxicity increases (figure 2). This group therefore has a high unmet need for new, less toxic treatments and improved treatment delivery approaches.

## Existing evidence: induction therapy
### Treatment

Lenalidomide is an immunomodulatory agent and thalidomide derivative available as an oral preparation, which is more potent in vitro and with a different adverse effect profile than thalidomide. For example, a major benefit of lenalidomide is the absence of associated neurotoxicity or sedation seen with thalidomide, making it more tolerable; however, there is a significant rate of myelosuppression (20%), which is not seen with thalidomide. Lenalidomide is licensed, in combination with dexamethasone, for use in newly diagnosed patients with multiple myeloma who are not eligible for transplant (TNE).

With the caveats of cross-trial comparison, our recent analysis of the Myeloma XI trial suggests that the combination cyclophosphamide, lenalidomide and attenuated dexamethasone (CRDa) is not superior to Rd-continuous used in the FIRST study (Study to Determine Efficacy and Safety of Lenalidomide Plus Low-dose Dexamethasone Versus Melphalan, Prednisone, Thalidomide in Patients With Previously Untreated Multiple Myeloma, NCT00689936) in terms of progression-free survival (PFS). In addition, Myeloma XI also showed that patients in the non-intensive pathway with only a minimal or partial response to induction therapy who were randomised to receive cyclophosphamide, bortezomib and dexamethasone consolidation had better outcomes than those who did not (an increase in median PFS of 11 months

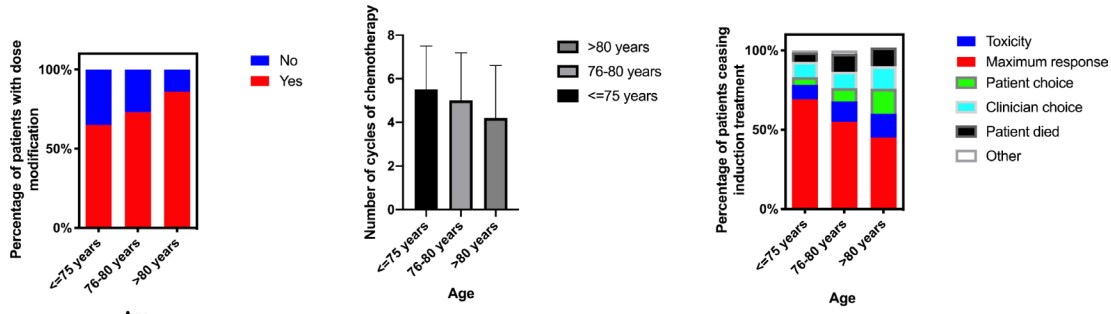

**Figure 2** Reasons for ceasing induction treatment in NCRI-XI (n=928). NCRI-XI, National Cancer Research Institute Myeloma XI Trial (ISRCTN68454111).

(HR: 0.73; p=0.061)).[4] However, a significant proportion of patients (394/610; 64.6%) did not make it to this randomisation due to participant withdrawal (211; 53.6% of 394), death (96; 24.4%), progression (45; 11.4%) and other reasons (42; 10.7%). These data suggest that combining IMiD agents and PI upfront may improve outcomes further, avoiding the early loss of patients and leading to improved responses, as we have seen with the carfilzomib, cyclophosphamide, lenalidomide and dexamethasone arm in the transplant-eligible pathway of the Myeloma XI study.[5 6]

Ixazomib (MLN9708) is an orally bioavailable, small molecule inhibitor of the 20S proteasome, and has shown single-agent activity in phase I/II studies alongside combination therapy with dexamethasone and more recently IMiD agents and alkylating agents.[7–9] The oral formulation provides convenience for patients, and the slower pharmacokinetic profile reduces the neurotoxicity seen with bortezomib, suggesting the use of this new PI in combination regimens might be better tolerated.[10] In the front-line setting, the combination of ixazomib with lenalidomide and dexamethasone (IRD) has been reported in phase I/II studies, demonstrating high response rates and good tolerability.[11]

With a favourable toxicity profile compared with either carfilzomib or bortezomib, and the benefits of oral dosing, the IRD combination represents a tolerable regimen to achieve the same combination of IMiD agents and PI in TNE patients. The Tourmaline MM-1 study demonstrated excellent efficacy and tolerability of IRD in the relapsed setting in TNE patients, supporting this hypothesis.[12] The Tourmaline MM-02 study, a randomised, double-blind, placebo-controlled study evaluating IRd versus placebo Rd has recently been reported[13] and showed a clinically meaningful but non-significant 13.5 month improvement in PFS in the IRd group (35.3 vs 21.8 months; HR 0.83; p=0.073). This trial used reactive dosing strategies.

## Treatment delivery

The International Myeloma Working Group (IMWG) proposed a scoring system for patient with myeloma frailty that predicts survival, adverse events (AEs) and treatment tolerability,[14] which can help to account for the considerable heterogeneity in outcome for TNE patients. This score combines age and the outcomes of three patient assessment tools; the Katz Activity of Daily Living,[15] Lawton's Instrumental Activity of Daily Living[16] and the Charlson Comorbidity Index[17 18] to categorise patients into three groups: fit, unfit and frail. The IMWG frailty score was subsequently shown to be predictive of both PFS and toxicity. An increase in frailty score was associated with an increased risk of death, progression, non-haematological AEs and treatment discontinuation that was independent of classical definitions of risk, including ISS stage and cytogenetic risk, and also independent of treatment regimen. As such, it was suggested to be useful in determining the feasibility of treatment regimens and appropriate dose reductions, but this remains to be prospectively validated.

## Existing evidence: maintenance therapy

Four published studies have demonstrated an important clinical benefit for the use of maintenance lenalidomide in newly diagnosed multiple myeloma in patients of all ages. Data from our previous study Myeloma XI contributes to this evidence base and in TNE patients, lenalidomide maintenance demonstrated a significant improvement in PFS compared with observation, of 24 months versus 11 months from maintenance randomisation.[19] This improvement was seen across all subgroups of patients with multiple myeloma, including all cytogenetic risk groups and at all ages. OS data demonstrated a benefit for lenalidomide once the effect of subsequent therapies have been taken into account.[20]

Overall, the data for maintenance lenalidomide until disease progression in patients not eligible for stem cell transplant suggest that there is a clear and significant improvement in PFS and a possible OS benefit. The crucial question to answer, going forward, is whether the results seen with lenalidomide as a single agent for maintenance can be enhanced further by the use of a combination regimen.

The use of ixazomib monotherapy in the maintenance setting demonstrated efficacy and tolerability in previously untreated patients[21] and this has recently been confirmed in the phase III Tourmaline-MM4 trial in the non-transplant-eligible setting.[22] Adding ixazomib to lenalidomide maintenance has not been studied in a randomised phase III study.

## Existing evidence: summary

The recently developed, less toxic PI ixazomib[11] with novel IMiD agents/PI combinations for induction and maintenance treatment needs to be evaluated in the context of the IMWG frailty score where frailty-adjusted dosing was recommended,[14] but emphasised the need for prospective validation of their approach.

## Aims and objectives

The Frailty-adjusted therapy in Transplant Non-Eligible patients with newly diagnosed Multiple Myeloma (FiTNEss) trial aims to improve outcomes for TNE patients by investigating whether using prospective dose adjustments dependent on patient frailty will improve patients' ability to remain on therapy, reduce toxicity, and improve outcomes from randomisation 1 (R1). The trial also aims, from randomisation 2 (R2), to investigate whether doublet maintenance therapy improve outcomes compared with single-agent lenalidomide without prohibitive toxicity.

## Trial design

The FiTNEss trial is a phase III, multicentre, randomised, parallel group trial in newly diagnosed patients with MM, who are assessed to be TNE by their treating clinician. Following R2, the trial is also double-blind placebo-controlled with the participant and treating clinician blind to treatment allocation. The following report details

**Box 1    Randomisation 1: inclusion and exclusion criteria**

**Inclusion criteria:**

1. Newly diagnosed as having MM according to the updated International Myeloma Working Group diagnostic criteria 2014 requiring treatment.
2. Not eligible for stem cell transplant.
3. Aged at least 18 years.
4. Meet all of the following blood criteria within 14 days before R1:

**Haematological:**

a. Absolute neutrophil count (ANC)$\geq 1 \times 10^9$/L. Unless the participant has a known/suspected diagnosis of familial or racial neutropenia in which case an ANC$\geq 0.75 \times 10^9$/L is allowed. The use of growth factor support is permitted.
b. Platelet count$\geq 50 \times 10^9$/L, or, in the case of heavy bone marrow infiltration ($\geq 50$%) which in the opinion of the investigator is the cause of the thrombocytopenia and provided appropriate supportive measures and patient monitoring are in place, platelet count$\geq 30 \times 10^9$/L is permitted. Please note: Platelet transfusions are not allowed $\leq 3$ days prior to randomisation in order to meet these values.
c. Haemoglobin$\geq 80$ g/L. The use of red blood cell transfusions is permitted.

**Biochemical:**

d. Total bilirubin$\leq 3\times$ upper limit of normal (ULN).
e. Alanine aminotransferase and/or aspartate aminotransferase $\geq 3$ x ULN.
5. Meet the pregnancy prevention requirements:

**Female participants who:**

a. Are not of childbearing potential, OR
b. If they are of childbearing potential, agree to practice two effective methods of contraception, at the same time, from the time of signing the informed consent form until 90 days after the last dose of study drug, OR
c. Agree to practice true abstinence when this is in line with the preferred and usual lifestyle of the subject. (Periodic abstinence (eg, calendar, ovulation, symptothermal, postovulation methods) and withdrawal are not acceptable methods of contraception).

**Male participants, even if surgically sterilised (ie, status post vasectomy), must agree to one of the following:**

a. Agree to practice effective barrier contraception during the entire study treatment period and through 90 days after the last dose of study drug, OR
b. Agree to practice true abstinence when this is in line with the preferred and usual lifestyle of the subject. (Periodic abstinence (eg, calendar, ovulation, symptothermal, postovulation methods) and withdrawal are not acceptable methods of contraception).

Contraception for female and male participants must be in accordance with (and participants must consent to) the Celgene-approved Pregnancy Prevention Programme.
If female and of childbearing potential, they must have a negative pregnancy test performed by a healthcare professional in accordance with the Celgene Pregnancy Prevention Programme.
6. Able to provide written informed consent.

**Exclusion criteria:**

1. Smouldering MM, monoclonal gammopathy of unknown significance (MGUS), solitary plasmacytoma of bone or extramedullary plasmacytoma (without evidence of MM).

*Continued*

**Box 1    Continued**

2. Received previous treatment for MM, with the exception of local radiotherapy to relieve bone pain or spinal cord compression, prior bisphosphonate treatment, or corticosteroids as long as the total dose does not exceed the equivalent of 160 mg dexamethasone.
3. Known resistance, intolerance or sensitivity to any component of the planned therapies.
4. Prior or concurrent invasive malignancies except the following:
   – Adequately treated basal cell or squamous cell skin cancer.
   – Incidental finding of low grade (Gleason 3+3 or less) prostate cancer requiring no intervention.
   – Adequately treated carcinoma in situ of the breast or cervix no longer requiring medical or surgical intervention.
   – Any cancer from which the subject has been disease free for at least 3 years.
5. Pregnant, lactating or breastfeeding female participants.
6. Major surgery within 14 days before randomisation. This would include surgical intervention for relief of cord compression but does not include vertebroplasty or kyphoplasty.
7. Systemic treatment, within 14 days before the first dose of ixazomib with strong CYP3A inducers (eg, rifampicin, rifabutin, carbamazepine, phenytoin, phenobarbital), or use of St. John's wort.
8. Any concomitant drug therapy which, in the opinion of the investigator, may lead to an unacceptable interaction with any of the agents ixazomib, lenalidomide, dexamethasone, and that cannot be safely stopped prior to trial entry. Full details of interactions can be found in the Summary of Product Characteristics.
9. Known gastrointestinal (GI) disease or GI procedure that could interfere with the oral absorption or tolerance of trial treatment, including difficulty swallowing.
10. $\geq$Grade 2 peripheral neuropathy.
11. Known HIV positive.
12. Participant has current or prior hepatitis B surface antigen positive or hepatitis C antibody positive. Participants must have screening conducted within 14 days before R1.
13. Active systemic infection.
14. Any other medical or psychiatric condition which, in the opinion of the investigator, contraindicates the participant's participation in this study.
15. Receipt of live vaccination within 30 days prior to R1.

the trial protocol and follows the structure of the SPIRIT statement.[23] The SPIRIT checklist[24] can be found within the online supplemental material.

## METHODS
### Setting

The trial will be conducted at 87 centres around the UK (see online supplemental material), as identified via a feasibility assessment to determine the most appropriate to participate in the trial. The majority of potential participants will be identified by the research team at the time they are referred to the haematology outpatient department with suspected multiple myeloma. A smaller number of participants may be identified during inpatient admissions. Invitation to participate in the trial and provision of information will be made either

during their first consultation, when routine diagnostic tests will be performed and potential treatment options discussed, or at the time they receive their diagnostic test results.

## Eligibility criteria

Adults (18 years and older) with newly diagnosed MM, by IMWG 2014 diagnostic criteria,[25] who are TNE and who are capable of giving written informed consent will be assessed for eligibility. Eligibility will be confirmed prior to each randomisation by the principal investigator or authorised delegate and will be recorded in the participant's medical records and on the relevant case report form (CRF). The participant will be registered into the trial prior to undergoing procedures that are specifically for the purposes of the trial and are above National Health Service standard of care.

To be eligible for R1, participants must meet all the inclusion criteria and none of the exclusion criteria outlined in box 1. Following 12 cycles of induction therapy, participants who achieve at least a minimal response (MR), according to IMWG uniform response criteria, and fulfil all the inclusion criteria and none of the exclusion criteria outlined in box 2, will proceed to R2.

## Interventions and dosing
### Intervention schedule

The control and experimental interventions for induction (R1) and maintenance (R2) therapy include: lenalidomide, ixazomib, placebo and dexamethasone. At R1, eligible patients will be allocated to one of two interventions with IRD induction; standard up-front dosing followed by toxicity dependent dose modification (reactive), or; frailty score-adjusted up-front dose reductions (adaptive). At R2, eligible patients will be randomised between lenalidomide+ixazomib (R+I) or lenalidomide and placebo maintenance therapy. Table 1 summarises the dosing schedules for R1 and R2. Information regarding dosing due to liver and renal function is provided in online supplemental material.

As part of their induction therapy, participants in the adaptive arm of R1 will have their dose adjusted according to changes in frailty category at the start of cycles 3, 5 and 7 (For the Myeloma Frailty index, participant's age is at the time of (Main) Trial registration, therefore, a patient's frailty will never change based on age only). Doses can also be escalated for suboptimal responders under certain criteria. If after cycle 2, a participant on the unfit or frail dosing strategy has not achieved at least an MR, or required a dose reduction due to toxicity, a request can be made to increase any of their doses to the next highest level at the start of cycle 4. A similar request can be made after cycle 4 for the start of cycle 6, but this requires at least a partial response. These criteria apply for all participants, irrespective of changes in frailty at cycles 3 and 5.

---

### Box 2   Randomisation 2: inclusion and exclusion criteria

**R2 inclusion criteria:**
1. Randomised into the Frailty-adjusted therapy in Transplant Non-Eligible patients with newly diagnosed Multiple Myeloma (Myeloma XIV) trial and received induction chemotherapy with ixazomib and lenalidomide continued for 12 cycles.
2. Achieved at least minimal response at the end of lenalidomide and dexamethasone induction according to the International Myeloma Working Group (IMWG) Uniform Response Criteria for Multiple Myeloma, with no evidence of progression prior to R2.
3. Meet all of the following blood criteria within 14 days before R2:

**Haematological:**
a. Absolute neutrophil count (ANC)$\geq 1 \times 10^9$/L. Unless the participant has a known/suspected diagnosis of familial or racial neutropenia in which case an ANC$\geq 0.75 \times 10^9$/L is allowed. The use of growth factor support is permitted.
b. Platelet count$\geq 50 \times 10^9$/L. Please note: Platelet transfusions are not allowed $\leq 3$ days prior to randomisation in order to meet these values.
c. Haemoglobin$\geq 80$ g/L. The use of red blood cell transfusions is permitted.

**Biochemical:**
d. Total bilirubin$\leq 3\times$ upper limit of normal (ULN).
e. Alanine aminotransferase and/or aspartate aminotransferase $\geq 3$ x ULN.

**R2 exclusion criteria:**
1. Received any antimyeloma therapy other than their randomised trial treatment, with the exception of local radiotherapy to relieve bone pain (in the absence of disease progression), or bisphosphonate treatment.
2. SD or disease progression according to the IMWG Uniform Response Criteria for Multiple Myeloma.
3. Known resistance, intolerance or sensitivity to ixazomib or lenalidomide that required cessation of either agent during induction.
4. Developed any malignancy since R1 except the following:
   – Adequately treated basal cell or squamous cell skin cancer.
   – Incidental finding of low grade (Gleason 3+3 or less) prostate cancer requiring no intervention.
   – Adequately treated carcinoma in situ of the breast or cervix no longer requiring medical or surgical intervention.
5. Pregnant, lactating or breastfeeding female participants.
6. Major surgery within 14 days before randomisation. This does not include vertebroplasty or kyphoplasty.
7. Systemic treatment, within 14 days before the first dose of ixazomib with strong CYP3A inducers (eg, rifampicin, rifabutin, carbamazepine, phenytoin, phenobarbital), or use of St. John's wort.
8. Known gastrointestinal (GI) disease or GI procedure that could interfere with the oral absorption or tolerance of trial treatment, including difficulty swallowing.
9. $\geq$Grade 2 peripheral neuropathy, or grade 1 with pain.
10. Known HIV positive.
11. Current or known hepatitis B surface antigen positive or hepatitis C antibody positive.
12. Active systemic infection.
13. Any other medical or psychiatric condition which, in the opinion of the investigator, contraindicates the participant's continued participation in this study.
14. Receipt of live vaccination within 30 days prior to R1 or receipt of live vaccination at any point during the trial prior to R2.

---

**Table 1** Dosing schedule

**Randomisation 1**

| Treatment | Induction—FIT+induction—standard dosing | Induction—unfit | Induction—frail |
|---|---|---|---|
| Lenalidomide (days=1–21) | 25 mg | 15 mg | 10 mg |
| Ixazomib (days=1, 8, 15)* | 4 mg | 4 mg | 4 mg |
| Dexamethasone †(days=1, 8, 15, 22) | 40 mg in participants≤75 years<br>20 mg in participants>75 years | 20 mg | 10 mg |

**Randomisation 2**

| Treatment | Lenalidomide+placebo maintenance | Lenalidomide+ixazomib maintenance |
|---|---|---|
| Lenalidomide (days=1–21) | 10 mg† | 10 mg† |
| Ixazomib (days=1, 8, 15)* | N/A | 4 mg† |
| Placebo (days=1, 8, 15) | 4 mg† | N/A |

*Ixazomib was not used in general multiple myeloma practice at the time of the European Myeloma Network publication. The following licensed dose of Ixazomib will be used for both randomisation arms: 4 mg, at days 1, 8 and 15. This has been studied in patients who are not eligible for transplant and was well tolerated. There have been no studies examining lower doses of Ixazomib so dose reductions are not permitted out of concern for loss of efficacy. The same dose is used irrespective of frailty.
†Or final dose administrated at the end of induction treatment if lower.

### Intervention adherence

Throughout the trial, lenalidomide and ixazomib will be taken orally and swallowed whole at the same time on the scheduled days. Dexamethasone will be administrated in accordance with the relevant Summary of Product Characteristics (SPCs). To monitor treatment adherence, participants will complete a daily medication diary, which will be reviewed at trial visits. Unused capsules will be returned to pharmacy.

### Dose modification and discontinuation

Both R1 and R2 treatment cycles will be 28 days in length. Response will be assessed at the end of each cycle according to the IMWG 2016 Uniform Response Criteria.[26 27] In the absence of progression or treatment intolerance, participants will receive a maximum of 12 cycles of induction therapy and continuous maintenance therapy. Those who receive 12 cycles of induction therapy will be assessed for R2 eligibility, with non-eligible participants being treated off trial.

Toxicity and hence treatment intolerance will be assessed throughout each treatment cycle, according to the National Cancer Institute (NCI) common terminology criteria for AE (CTCAE) V.5.

For a new cycle of treatment to begin (induction and maintenance), the participant must meet the haematological and biochemical criteria (at day 1 or ≤3 days prior) outlined in the eligibility criteria (R1: box 1, R2: box 2). Non-haematological toxicities (except for alopecia) must have resolved to less than or equal to grade 1 or to the participants' baseline condition in order for treatment to resume. If the participant does not meet these criteria, their dose will be delayed for 1 week before they are reassessed. This will continue for a maximum of 3 weeks before treatment discontinuation or 8 weeks at the discretion of the chief investigator.

In the event that a dose is reduced due to toxicity, as per the lenalidomide SPC the dose may be reintroduced to the next higher dose level on improvement. Ixazomib, once reduced cannot be re-escalated.

### Concomitant medication

Concomitant medication, disease and other malignancies will be recorded at eligibility.

All participants may receive additional care during the treatment period as deemed appropriate by the treating clinician. Local support care protocols, including anti-emetic schedules, tumour lysis syndrome prevention, venous thromboembolism prophylaxis and prophylactic antimicrobial therapy, will be followed for both randomisations. Permitted and excluded concomitant medications and procedures can be found in the online supplemental material.

### Outcomes
#### Primary outcome

For R1, the primary outcome is early treatment cessation (defined as a binary endpoint) for reactive versus adaptive dosing in participants defined to be 'unfit' or 'frail' at baseline. Participants will be defined to have experienced an event if they die, progress, or are withdrawn from treatment (by the treating clinician) or withdraw consent for treatment, within 60 days of R1.

For R2, the primary outcome is PFS for R+placebo versus R+I, and is defined as the time from R2 to the time of first documented disease progression or death from any cause. Individuals lost to follow-up or progression free at time of analysis will be censored at their last known alive and progression-free date.

#### Secondary outcomes

The secondary outcomes of this trial are to assess PFS for reactive versus adaptive dosing, time to progression, time

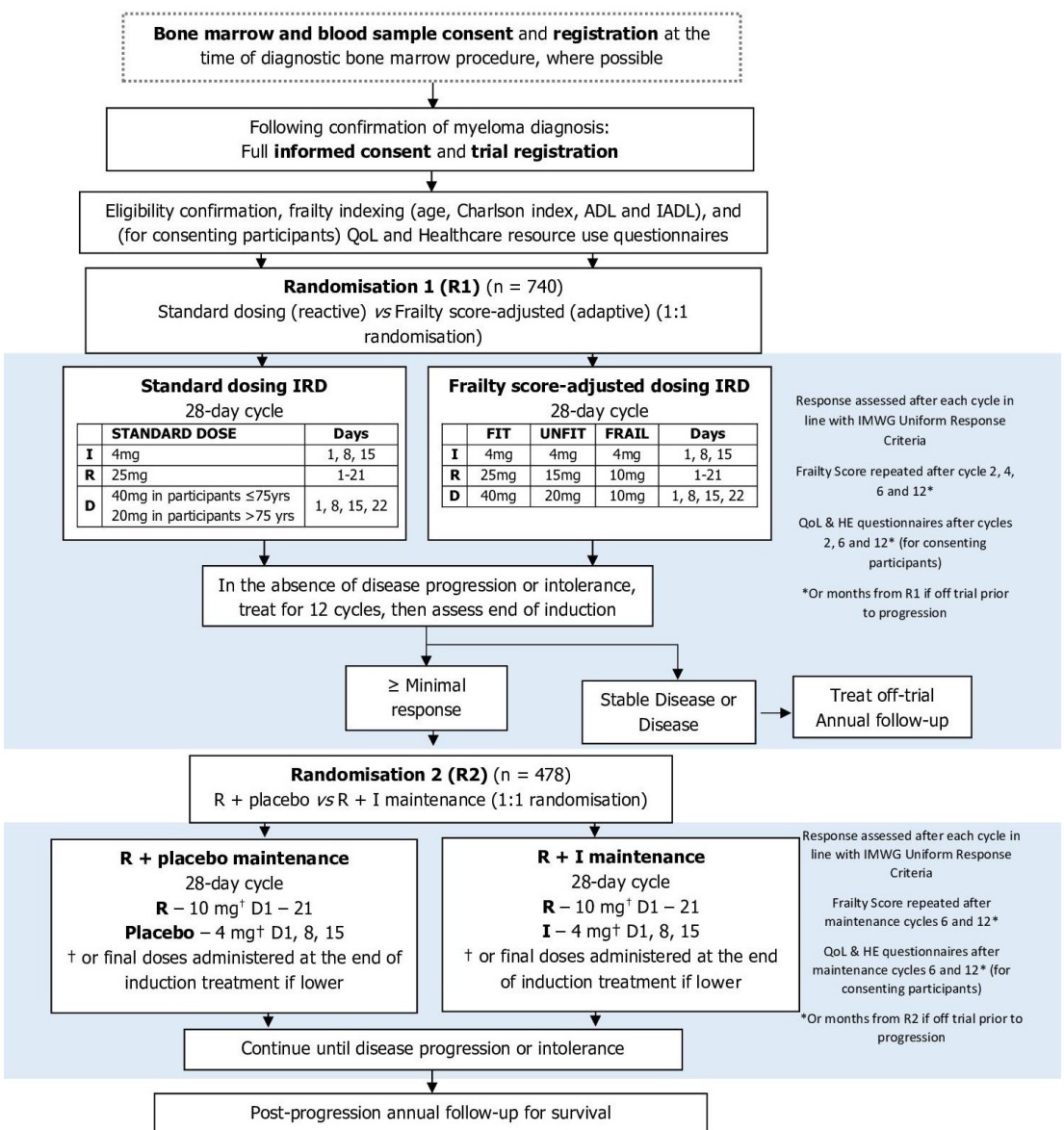

**Figure 3** Flow diagram of Myeloma XIV (Frailty-adjusted therapy in Transplant Non-Eligible patients with newly diagnosed Multiple Myeloma) Trial. ADL, activity of daily living; HE, health economics; IADL, instrumental activity of daily living; IMWG, International Myeloma Working Group; IRD, ixazomib, lenalidomide and dexamethasone; QoL, quality of life; R+I, lenalidomide+ixazomib.

to 2nd PFS event, OS, survival after progression, deaths within 12 months of R1, overall response rate, attainment of ≥ very good partial response (VGPR), attainment of minimal residual disease (MRD) negativity (flow MRD will be assessed only. Additional detail on the testing of MRD and timepoints is presented in the 'central lab analysis' section of the online supplemental material), duration of response, time to improved response, time to next treatment, treatment compliance and total amount of therapy delivered, toxicity and safety including the incidence of second primary malignancies, quality of life (QoL), cost-effectiveness of reactive versus adaptive dosing of IRD and cost-effectiveness of R+I versus R.

### Exploratory outcomes
Exploratory outcomes are to prospectively validate the UK Myeloma Research Alliance (UK-MRA) Myeloma Risk

Profile, to assess the usefulness of the Karnofsky performance status, and consider the association of molecular subgroups with response, PFS and OS.

### Participants timelines
The full trial schema can be seen in figure 3. The schedule of assessments at each timepoint is presented in figure 4.

### Trial entry
Participants will enter the trial at one of two points in their patient pathway, this will either at bone marrow registration or main trial registration.

### Trial consent
Participants who enter the trial at bone marrow registration will provide consent to having bone marrow and blood samples taken and sent to central laboratories for

## Local Investigations

| Investigation | Consent & trial registration | Pre-R1 (assessments for eligibility within 14 days prior to R1, unless otherwise stated) | Day 1 (or ≤3 days prior) of each induction treatment cycle | End of 12 cycles of IRD induction treatment | Pre-Randomisation 2 (R2) (assessments for eligibility within 14 days prior to R2, do not need repeating if end of IRD assessments are within this timeframe) | Day 1 (or ≤3 days prior) of each maintenance treatment cycle | Disease Progression |
|---|---|---|---|---|---|---|---|
| Pre-trial Consent for Bone Marrow (at time of diagnostic bone marrow) and bone marrow registration | ✓ | | | | | | |
| Full written informed consent and (main) trial registration | ✓ | | | | | | |
| Medical history (including comorbidities, concomitant medications and previous malignancies) | | ✓ | | | | | |
| Assessment of cardiac and thyroid function (as part of standard care) | Monitor throughout treatment | | | | | | |
| Physical examination (including height/weight, BP, performance status*, vital signs [baseline only]) | | ✓ | ✓ | ✓ | ✓ | ✓ | ✓ |
| Pregnancy test (if female of child bearing potential, see Appendix 8) | | ✓ | ✓ | ✓ | ✓ | ✓ᵇ | ✓ |
| FBC, U&Es, calcium, creatinine, LFTs (including bilirubin, and AST and/or ALT), albumin, LDH, calculated creatinine clearance, urinary protein:creatinine ratio* *only requested at baseline | | ✓ | ✓ | ✓ | ✓ | ✓ | ✓ |
| CRP, β2M | | ✓ (After cycles 2, 4, and 6 d) | ✓ (After cycle 2, 4, and 6 d) | ✓ | | ✓ (After cycles 6 and 12 c) | ✓ |
| Serum paraprotein, serum free light chains, serum total (class-specific) immunoglobulins and urinary light chain detection (quantification where available) | | ✓ (To assess response to previous cycle, not applicable for C1) | ✓ (To assess response to previous cycle, not applicable for C1) | ✓ | ✓ | ✓ (To assess response to previous cycle) | ✓ |
| Bone marrow aspirate and (if available) trephine | | ✓ (Within 6 weeks prior to R1) g | If at any point a first occurrence of CR or sCR is suspected then bone marrow aspirate should be sent for local review as well as to HMDS, as detailed below. (s)CR cannot be confirmed without bone marrow. | | | | ✓ |
| Serology of hepatitis B and C | | ✓ | | | | | |
| Imaging | | ✓ (Within 3 months prior to randomisation) | Imaging of lytic and/or focal bone and extramedullary lesions if clinically indicated, in accordance with IMWG recommendations and local practice * | | | | |
| IMWG Frailty Index (Charlson Comorbidity Score, IADL, ADL) | | ✓ | ✓ (After cycles 2, 4, and 6 d) | ✓ | | ✓ (After cycles 6 and 12 c) | ✓ |
| Quality of Life and Healthcare resource use questionnaires (EORTC QLQ-C30, QLQ-MY20, EQ-5D (3 Level)) *Completed by consenting participants in clinic | | ✓ (Before participant is informed of allocated dosing strategy) | ✓ (After cycle 2 and 6 d) | ✓ᶜ | | ✓ (After cycles 6 and 12 c) | ✓ |
| Adverse Events | | | Monitor throughout study and report on relevant eCRFs (from randomisation until 60 days post last treatment dose) | | | | |
| SAEs / SUSARs / SPMs | | | Monitor throughout study, all SUSARs & SAEs/SPMs must be reported to CTRU within 24 hours of the Site becoming aware of the event: refer to Section 14: (SAEs from randomisation until 60 days post last treatment dose & SPMs/SUSARs/SUSARs from first treatment dose until the end of trial) | | | | |

## Central Analysis Investigations: Central Samples (all Participants – core consent)

| Sample | Send to | Investigation | Bone marrow and blood sample consent samples only | Pre-randomisation 1 (pre-treatment) – Post Main Consent | During induction therapy | At the end of 12 cycles of IRD induction treatment (or at end of the final IRD cycle if sooner) | During maintenance therapy | Disease Progression |
|---|---|---|---|---|---|---|---|---|
| 20 mL whole clotted blood or 10 mL serum | Birmingham | To confirm disease response & progression through investigation of paraprotein, immunoglobulins, serum free light chain, urinary free light chain and β2M | | ✓ | ✓ (After 2 cycles of IRD) | ✓ | ✓ (At 2 monthly intervals) | ✓ |
| 10 mL random urine | | | | ✓ | ✓ (After 2 cycles of IRD) | ✓ | ✓ (At 2 monthly intervals) | ✓ |
| 2 mL bone marrow aspirate in EDTA | HMDS, Leeds | To determine MRD | ✓ (Within 8 weeks prior to R1) | ✓ᵈ | ✓ (After 6 cycles of IRD) | ✓ | ✓ (At 12 months post-maintenance randomisation (R2)) | ✓ |
| 3 mL bone marrow aspirate in EDTA | ICR, London | Cytogenetic/Molecular research (including MLPA / FISH) | ✓ (Within 8 weeks prior to R1) | ✓ᵈ | If at any point a first occurrence of CR or sCR is suspected then bone marrow aspirate should be sent to HMDS. | | | |
| 6 mL peripheral blood in EDTA | | | ✓ (Within 8 weeks prior to R1) | ✓ᵈ | | | | |
| 20 mL peripheral blood in EDTA | LIMR, Leeds | Frailty biomarker studies | | ✓ | ✓ (After cycle 2, 4 & 6) | ✓ | | ✓ |
| 10 mL clotted blood | | | | ✓ | ✓ (After cycle 2, 4 & 6) | ✓ | | ✓ |
| 2 mL bone marrow aspirate in EDTA | | | | | | ✓** | ✓ (At 12 months post-maintenance randomisation (R2)) | ✓ |

*Figure caption is on the next page*

**Figure 4** Summary of investigations (local and central). ADL, activity of daily living; ALT, alanine aminotransferase; AST, aspartate aminotransferase; BP, blood pressure; CR, complete response; CRP, C-reactive protein; CTRU, Clinical Trial Research Unit; eCRFs, electronic case report forms; EDTA, edetic acid; EORTC QLQ C30, European Organisation for Research and Treatment of cancer quality of life questionaire; EQ-5D, Euroqol 5 dimensions; FBC, full blood count; FISH, fluorescence in situ hybridization; HMDS, Haematology Malignancy Diagnostic Service; IADL, instrumental activity of daily living; ICR, Institute of Cancer Research; IMWG, International Myeloma Working Group; IRD, ixazomib, lenalidomide and dexamethasone; LDH, lactate dehydrogenase; LFTs, liver function test; LIMR, Leeds Institute of Medical Research; MRD, minimal residual disease; QLQ-MY20, myeloma quality of life questionaire; SAEs, serious adverse events; sCR, stringent complete response; SPMs, secondary primary malignancies; SUSARs, suspected unexpected serious adverse reactions; U&E's, urea and electrolytes. **a.** The FBC should be repeated mid-cycle 1 (Day 14 +/−3 days) or more frequently and during subsequent cycles if there is a concern about cytopenias, **b.** Pregnancy test must also be performed at 4 weeks after the end of study treatment, **c.** Or at 6 and 12 months post-R2 if treatment is stopped prior to this for reasons other than disease progression, **d.** Or at 2, 6 and 12 months post-R1 if treatment is stopped prior to this for reasons other than disease progression, **e.** The imaging at baseline/pre-registration is mandatory. Subsequent imaging will be as per local protocols/standard of care – imaging will only need to be repeated if extramedullary disease was detected at baseline, or in the event of new symptoms suggestive of new extramedullary disease, cord compression, new fracture, etc, or to investigate new hypercalcaemia. In the rare event of participants with extramedullary disease at baseline, imaging should be performed at the end of IRD induction to confirm the end of induction response, and at any other time the disease parameters suggest that the participant has achieved a complete response (if not already achieved at the end of induction) to confirm that the extramedullary disease has resolved, **f.** Performance status should be recorded at the same timepoints as the frailty index (ie, prior to R1, after cycles 2, 4, 6 and 12 of induction, and after cycles 6 and 12 of maintenance). Performance status should also be recorded at the time of disease progression. Or at 2, 6 and 12 months post-R1 if treatment is stopped prior to this for reasons other than disease progression, **g.** If the participant did not consent to the pre-trial bone marrow registration part of Myeloma XIV, the bone marrow biopsy will need to be taken after full informed consent. *At pre-randomisation 1 a CD138 negative portion of the bone marrow aspirate will be sent to LIMR after CD138 positive selection at ICR, **At the end of induction bone marrow aspirate will be sent to ICR after CD138 positive selection at LIMR.

analysis. If the participant is diagnosed with a plasma cell dyscrasia, other than myeloma, or they have myeloma, but decide not to take part in the FiTNEss trial, they will also have the option of consenting to their samples being used in future research.

All participants will provide written informed consent for the trial prior to trial registration. Optional consent for QoL resources and the use of samples for future research will also be obtained.

### Trial registration
Following trial consent, participants will be registered onto the main trial and assessed for eligibility. Consenting patients will complete the baseline QoL and healthcare resource use questionnaires. Trial samples for blood and urine will be taken for all participants, and bone marrow samples will be taken for those who did not enter the trial through bone marrow registration.

### Trial treatment
Following trial registration, participants will be randomised into R1 and treated as described in the intervention schedule on a monthly basis. In absence of disease progression or intolerance, those participants with at least an MR following 12 cycles of maintenance and whom meet all of the R2 eligibility criteria will proceed to R2 maintenance treatment.

Participants will be followed up monthly (at each cycle) while receiving maintenance treatment, until death or until the final analysis of the trial (whichever happens sooner).

### Trial follow-up
Participants who discontinue treatment during induction and before the point of R2 will be followed up to the point of assessment for eligibility for R2, unless they withdraw consent for this. Thereafter, participants will continue to be followed-up for data pertaining to safety, progression (including second progression), and survival. Frailty scores will be completed for all participants at 2, 4, 6 and 12 months post R1, irrespective of whether they have discontinued treatment. Similarly, QoL and healthcare resources use questionnaires (if the participant has consented to these) will continue to be completed at 2, 6 and 12 months post R1.

If treatment has been stopped without progression following R2, for example, due to toxicity, then participants will be followed up 2 monthly until disease progression. Follow-up will include local investigations, central investigations, frailty index at 6 and 12 months post R2 and QoL and healthcare resource use at 6 and 12 months post R2 (if consented).

### Sample size
In total, 740 participants will be enrolled into the trial at R1 to ensure that at least 478 participants remain on trial and are randomised to R2. It is assumed that 65% of those randomised at R1 will be progression free and, therefore,

eligible for R2, hence 740 participants are required to be enrolled.

Based on data from the Myeloma XI non-intensive pathway, we hypothesise the frail and unfit patients in R1 will be similar to the older patients in Myeloma XI (>75 years) who have an early treatment cessation rate (within 60 days of R1) of 20%. This hypothesis is based on the expectation that the frailty score is heavily driven by patient age. Younger patients (≤75 years) in the Myeloma XI non-intensive pathway have an early treatment cessation rate of 9%, and it is our hypothesis that our frailty-based dosing schedule has the potential to reduce the rate among the unfit and frail patients to the proportion observed in fit patients.[28 29]

To demonstrate a decrease of 11% in the proportion of early treatment cessation from 20% in the standard dosing schedule arm to 9% in the frailty score-adjusted dose arm among those patients scored at baseline to be unfit or frail would require the recruitment of 324 patients with an allocation ratio of 1:1. These calculations are based on a Pearson's $\chi^2$ test without continuity correction, assume a two-sided 5% level of significance, 80% power, and allow for a 1% dropout prior to 60 days post randomisation. Given that we anticipate that 45% of patients will be scored as unfit or frail, by assuming this trial will have a similar underlying population as in Myeloma XI non-intensive pathway and the age distributions in the IMWG report proposing the frailty score,[14] we would anticipate that we will require 720 patients to enter the trial at R1 to have sufficient unfit and frail patients available to answer this question.

For R2, in the non-intensive pathway of Myeloma XI, the median PFS for patients on R following CRDa induction was approximately 33 months,[19] where approximately 65% of individuals were progression-free 12 months post randomisation. As R2 is approximately 12 months following R1 in FiTNEss, we assume that the median PFS for patients receiving R maintenance therapy will be 21 months from R2. Tourmaline-MM1[12] demonstrated an HR for PFS of 0.74 when comparing IRD and RD in patients with relapsed and refractory multiple myeloma. Thus a similar HR would be the minimum clinically relevant difference for our comparison in patients with newly diagnosed multiple myeloma. Assuming a median PFS of 29 months for those in the R+I maintenance group in addition to the assumption of 21 months in the R group equates to an HR of 0.72.

The above assumptions require the recruitment of 478 participants over a 30 month recruitment period with a further 24 months of follow-up. These calculations also assume a two-sided 5% significance level, 80% power and allow for a 3% dropout rate prior to a PFS event being experienced. Note that 80% power is attained when 302 events have been observed. A total of 740 participants should be allocated to R1 to ensure that 478 participants are available at the second randomisation.

OS is considered to be a key secondary endpoint for R2, 180 events with a minimum of 2-year follow-up for all participants are required for 80% power.

## Recruitment

It is planned that 740 participants will be recruited over a 30-month recruitment period from 87 UK centres. Once all centres are open, the recruitment target is 30 participants a month. The trial opened to recruitment on 4 August 2020. As of May 2021, 85 sites are open to recruitment and 137 participants have been randomised to the first stage of the trial.

In order to ensure the trial will meet the target sample size within the recruitment period, site set-up was prioritised while the trial was preparing to open to recruitment. In addition, the trial team are actively engaging with principal investigators at sites who support the trial to ensure that those sites open quickly and are maintaining regular communication with open sites to ensure that they continue to recruit to the trial. Finally, while the trial was originally delayed due to the COVID-19 pandemic, efforts were made to ensure that it opened as soon as possible when research restarted. One factor being the positive risk to benefit ratio of the number of hospital visits required for the trial interventions as compared with standard of care.

## Assignment of treatment allocations

Each of the registration and randomisation procedures will be conducted centrally using the Leeds Clinical Trial Research Unit (CTRU) automated 24-hour web-based and telephone system.

## Registrations

If a patient is suspected to have myeloma, they will enter the trial at the time of routine diagnostic tests, through bone marrow registration. Once diagnosis is confirmed locally and the research team consider the patient potentially eligible for the trial, patients will be provided with full verbal explanation of the trial and the full participant information sheet and informed consent documents to consider. Once the participant has provided informed consent, they will be registered onto the trial.

Other participants who have myeloma confirmed prior to entering trial will enter at trial registration.

## Randomisations

Following trial registration, patients will be assessed for R1 eligibility. Eligible participants will be randomised on a 1:1 basis into R1, using the stratification factors; centre, IMWG frailty category, beta-2 microglobulin concentration (<3.5, 3.5 to <5.5, ≥5.5 mg/L), Haemaglobulin concentration (<100, ≥100 g/L, serum crestinine concentration (<175, ≥175 µmol/L), corrected serum calcium concentration (<2.75, ≥2.75 mmol/L) and platelets (<150, ≥150×10⁹/L).

Following R1, and as soon as the end of induction response is known, R2 eligible participants will be randomised on a 1:1 basis into R2, using the stratification

factors; centre, allocated induction arm (reactive, adaptive); final response to induction treatment (<VGPR, ≥ VGPR).

For both R1 and R2, a computer generated minimization programme that incorporates a random element will be used to ensure treatment groups are well balanced for the specified stratification factors.

### Blinding methods

For R2 treatment, allocation will be concealed from participants, treatment provider and the trial team. The placebo and ixazomib capsules will be identical in colour, size, packaging and labelling.

To maintain the overall integrity of the trial design, unblinding will only be permitted in exceptional circumstances; for example, valid medical or safety reasons where assuming that the patient is receiving active treatment and/or stopping the blinded medication is insufficient.

Unblinding will be conducted automatically using an online system accessed by an authorised member of the site research team.

If unblinding is performed at any stage during the trial, decisions around further trial treatment will be the responsibility of the principal investigator or delegate. In either case, unblinded participants will be followed up as per the protocol.

At the completion of the trial and after final analysis, participants will be given the opportunity to be informed of their allocation by their research site.

### Data collection

Clinical data will be collected both electronically and on paper by staff at the research site completing CRFs provided by CTRU. QoL and Healthcare Resource Use questionnaires will be completed on paper CRFs by the participant. All paper CRFs will be sent to CTRU, by the research site, usually via standard post and entered onto an electronic database. These data along with the data entered electronically by staff at each research sites will be validated and monitored for completeness and quality by the CTRU.

Missing data will be chased until it is received, confirmed as not available or the trial is at analysis. Missing QoL data items will not be chased from participants, although missing questionnaires may be chased from sites.

### Data management

Validation checks will be incorporated into the trial database to verify the data, and discrepancy reports will be generated for resolution by the trial site. Priority validations will be incorporated to ensure that any discrepancies related to participant rights, or the safety of participants, are expedited to sites for resolution. The CTRU/ Sponsor will reserve the right to intermittently conduct source data verification exercises on a sample of participants, which will be carried out by staff from the CTRU/ Sponsor. Source data verification will involve direct access to participant notes at the participating hospital sites and the ongoing central collection of copies of consent forms and other relevant investigation reports.

### Statistical methods

Statistical analysis is the responsibility of the CTRU statisticians, with the exception of the analysis for cost-effectiveness of delivery of IRD and R/R+I, which will be undertaken by Health Economists at the University of Leeds. A full statistical analysis plan and health economics analysis plan (HEAP) will be written and approved before any analysis is undertaken.

All analyses will be conducted on the intention-to-treat population, where participants will be included according to their randomisation allocation regardless of eligibility, whether they prematurely discontinued treatment, or did not comply with the regimen. A per-protocol analysis, where participants will be included if they received their allocated intervention according to the protocol, will be considered for the primary endpoints if there are a considerable number of major protocol violators. The safety population will consist of all participants who received at least one dose of the trial treatment and participants will be summarised as per their treatment received rather than their allocation.

An overall two-sided 5% significance level will be used for all efficacy endpoint comparisons. For the primary endpoints, this will be adjusted to account for the formal interim analyses.

#### Primary endpoint analysis
##### Randomisation 1

For R1, the number and proportion of participants, categorised as unfit or frail at baseline, experiencing an early treatment cessation event will be summarised by randomisation allocation and exact 95% CIs will be calculated.

A logistic regression model will regress early treatment cessation on randomisation allocation (reactive/adaptive dosing) adjusting for the stratification factors of the trial. A statistically significant induction treatment effect will be suggested if the p value for the resulting OR is <0.047. Parameter estimates, ORs and corresponding 95% CIs, df, test statistics and p values will be presented for each variable in the model. Residuals and predicted values produced from the models will be examined to assess the assumptions of the statistical models.

##### Randomisation 2

For R2, PFS between the two maintenance therapies (R+placebo/R+I) will be compared using a Cox regression model adjusting for the stratification factors of the trial. A statistically significant maintenance treatment effect will be suggested if the p value for the HR corresponding to randomisation allocation is <0.047. Parameter estimates, HRs and corresponding 95% CIs, df, test statistics and p values will be presented for each variable in the model.

The proportional hazards assumptions will be assessed by plotting the hazards over time (ie, the log cumulative

hazard plot) for each treatment arm and using appropriate statistical tests. If evidence is found to support the violation of the proportional hazards assumption, then alternative appropriate analysis methods will be investigated.

No imputation strategy is planned for the primary endpoints.

### Secondary endpoint analysis

Secondary endpoint analysis of OS and other time-to-event endpoints will be analysed using similar methods to those described for PFS.

MRD negativity and other binary endpoints will be analysed using similar methods to those described for the early treatment cessation primary endpoint. The number and proportion of participants in each response category (stringent complete response (sCR), complete response (CR), VGPR, etc) will be summarised by allocated treatment and exact 95% CIs will be calculated. The difference in proportions for each response category will be presented with corresponding 95% CIs.

The domains of the QoL questionnaires will be summarised using mean scores adjusted for baseline and 95% CIs at each assessment timepoint. Similar summaries will be produced for quality-adjusted life years (QALYs) derived using the EQ-5D-3L (Euroqol 5 dimensions) questionnaire.

### Exploratory and subgroup analyses

An overview of the planned exploratory and subgroup analysis can be found in the online supplemental material. These include genetic and molecular analysis of patient samples conducted by the respective central laboratories.

### Health economics

A full HEAP will be written and approved before any analysis is undertaken.

Economic evaluations will be conducted at R1 and R2, using within-trial and decision-model-based analyses. The analysis will be guided by the The National Institute for Health and Care Excellence (NICE) reference case, applying the cost-utility framework from the perspective of the health and social care provider over a life-time horizon. Base case QALYs will be based on EQ-5D-3L responses, and costs on patient completed resource use forms and hospital records. Results will be presented in terms of incremental cost-effectiveness ratios, cost-effectiveness acceptability frontiers and net benefit.

### Trial oversight

The trial management group (TMG) comprises of the chief and co-chief investigators, CTRU team and coinvestigators and are responsible for the clinical set-up, ongoing management, promotion of the trial and for the interpretation of the results. The trial steering committee (TSC), consisting of independent clinicians and statisticians, along with a patient representative, will provide overall supervision on the trial, including trial progress, adherence to protocol, participant safety and consideration of new information.

### Data monitoring

An independent data monitoring and ethics committee (DMEC) will review the safety and ethics of the trial by reviewing unblinded interim data prepared by the CTRU in strict confidence at approximately yearly intervals. Unblinded safety updates are also prepared at 6 monthly intervals. After each annual review of safety data, the DMEC will make their recommendations to the TSC about the continuation of the trial who will make their recommendations known to the TMG.

### Interim analyses

Two formal interim analyses will be undertaken for early efficacy, one for each of the randomisations. The first will occur when 50% of required participants (370 participants) have reached 60 days post R1. The second will occur when 50% of required PFS events have been observed (151 events) following R2. In order to maintain an overall two-sided 5% significance level for the primary endpoint analysis, the O'Brien and Fleming alpha spending function[30] will be used. This results in a 0.05% significance level for the interim analysis. The analysis itself will reflect that detailed in the statistical analysis section. For the second interim analysis, only the DMEC, safety statistician and supervising statistician will see the unblinded results, as is standard procedure for double-blind trials.

No other formal analysis of the trial is planned before the participants have attained the primary endpoints.

The DMEC, in the light of the interim data, and any advice or evidence they wish to request, will advise the TSC if there is proof beyond reasonable doubt that one treatment is better and recommend appropriate changes to the trial protocol.

### Harms
#### Adverse events

AEs are any untoward medical occurrence in a patient or clinical trial subject administrated a medicinal product and which does not necessarily have a causal relationship with this treatment. AEs can be defined as any unfavourable and unintended sign (including an abnormal laboratory finding, for example), symptom or disease. Due to the nature of myeloma and its treatment, participants are likely to experience several AEs throughout the course of the disease.

All AEs, both related and unrelated to myeloma treatment, will be collected on the relevant CRF from R1 until 60 days after the last dose of protocol treatment and will be evaluated and summarised in accordance with the NCI–CTCAE V.5.

#### Serious AEs

Serious AEs (SAEs) are defined as any untoward medical occurrences or effects that at any dose result in death; or are life-threatening (at the time of event); or require in patient hospitalisation or prolongation of existing

hospitalisation; or result in persistent or significant disability of incapacity; or result in a congenital abnormality or birth defect; or jeopardise the participant or may require an intervention to prevent one of the above outcomes/consequences (other important medical event). SAEs will be reported from R1 until 60 days post the last dose of trial drug.

Serious adverse reactions (SARs) are SAEs that are deemed to be possibly related to any dose administrated of any trial treatment. Suspected unexpected serious adverse reactions (SUSARs) are SARs, of which the nature and severity is not consistent with the applicable reference safety information. SUSARs and SARs will be reported from the date of the first trial drug for the duration of the trial.

### Presenting safety data
Safety analyses will summarise all SUSARs, SARs, SAEs, ARs, AEs and treatment-related mortality rates. Safety data will be presented by treatment group for the safety population in addition to suspected relationship to trial treatment.

### Secondary primary malignancies
All new secondary primary malignancies or suspected malignancies will be recorded from R1 for the duration of the trial and will be summarised and reviewed by an appointed member of the TMG, who will determine whether trial treatment should continue.

### Pregnancies
The Celgene approved pregnancy programme will be followed as per usual clinical practice. Pregnancies in participants on trial treatment will be prevented as effectively as possible. Pregnancies and suspected pregnancies in a female or male participant's partner occurring at any time until 90 days post cessation of trial treatment will be reported.

### Auditing
The CTRU and the trial Sponsor have procedures in place to ensure that serious breaches of GCP or the trial protocol are identified and reported. A triggered monitoring plan will ensure that sites at risk are monitored accordingly.

### Patient and public involvement
FiTNEss has been developed following extensive discussion within the UK myeloma community, including with the NCRI Myeloma Subgroup (UK-MRA) and the NCRI Haematological Oncology Group. Both groups include patient and public representatives who work with clinical members of the group. To develop studies which address key questions for induction and maintenance. The protocol was reviewed in depth by a patient representative in order to ensure that the interventions and proposed schedule of assessments would be acceptable to patients. In addition, the trial consent and participant information document was reviewed for clarity by the same patient representative. Furthermore, to ensure that the patient perspective is considered throughout the trial, a patient representative sits on the TSC.

## ETHICS AND DISSEMINATION
### Ethics approval statement
Ethical approval has been obtained from the North East—Tyne & Wear South Research Ethics Committee (reference 19/NE/0125). In addition, approval was granted by the appropriate local research and development department for each participating centre prior to opening to recruitment. Participants will be required to provide written informed consent before joining the trial.

### Protocol amendments
The trial opened to recruitment on 4 August 2020 using protocol V.2, dated 10 October 2019. An amendment to protocol V.3 is anticipated in June 2021, which will include the addition of the secondary endpoint event-free survival, clarification on the requirements around when face-to-face assessments should be conducted, and adding in the recommendation of 5 mg once daily of lenalidomide for participants with severe renal impairment, as opposed to 15 mg every other day.

### Consent
The principal investigator retains overall responsibility for the informed consent of participants at their site and must ensure that any medically qualified person delegated responsibility to participate in the informed consent process is duly authorised, trained and competent to participate according to ethical approved protocol, principles of Good Clinical Practice and Declaration of Helsinki. Written consent will be obtained and signed by a medically qualified member of the site research team. A record of the consent process for both bone marrow and blood sample consent and full trial consent, including the date of consent and all those present, will be kept in the participant's notes. At any stage, participants can withdraw consent without repercussion.

### Confidentiality
All information collected during the course of the trial will be kept strictly confidential. Information will be held securely on paper at Leeds CTRU. In addition, the CTRU will hold electronic information on all trial participants. The CTRU will have access to the entire database for monitoring, coordination, and analysis purposes.

### Access to data
Data will not be made available until the end of the study, The CTRU will control the final trial datasets, and any requests for data will be reviewed by the TMG in the first instance. Only requests that have a methodologically sound proposal and whose proposed use of the data has been approved by the independent TSC, based on the principles of a controlled access approach, and subject to existing contractual agreements with the funder, will be

considered. Proposals should be directed to Leeds Clinical Trials unit (CTRU-DataAccess@leeds.ac.uk) in the first instance; to gain access, data requestors will need to sign a data access agreement.

## Ancillary and post-trial care

Participants who stop trial treatment due to progression or any point prior to the end of trial will be treated off-trial at the discretion of their treating clinician. Following disease progression, participants will be followed up annually until death, or until the end of the trial for post-progression endpoints.

## Dissemination policy

Authorship of clinical and translational outputs will be in keeping with the UK-MRA Publication Policy and due acknowledgement to participants, local investigators, funders and NCRI Haematological Oncology Group support made. The success of the trial depends on the collaboration of all trial members. For this reason, credit for the main results will be given to all those who have collaborated in the trial, through authorship and contributorship. Uniform requirements for authorship for manuscripts submitted to medical journals will guide authorship decisions alongside the guidance of the UK-MRA.

To maintain the scientific integrity of the trial, data will not be released prior to the end of the trial or a primary endpoint being reached, either for trial publication or oral presentation purposes, without the permission of the TSC and the (co-)chief investigators. In addition, individual collaborators must not publish data concerning their participants that is directly relevant to the questions posed in the trial until the main results of the trial have been published and following written consent from the Sponsor.

## Appendices
### Informed consent material
The consent forms that are to be completed by the participant at bone marrow registration and/or trial registration are included in online supplemental material.

### Biological specimens
The collection of central samples for laboratory analysis is summarised in figure 4. The analysis to be conducted for trial purposes is stated in the online supplemental material. Additional analysis may be carried out by each central laboratory provided the appropriate consent for sample use in future research has been provided by the participant at trial entry.

**Author affiliations**
[1]Leeds Institute of Clinical Trials Research, University of Leeds Clinical Trials Research Unit, Leeds, UK
[2]Cancer Research UK London Research Institute, London, UK
[3]Department of Haematology, University Hospitals Bristol and Weston NHS Foundation Trust, Bristol, UK
[4]Department of Haematology, Kings College Hospital NHS Foundation Trust, Princess Royal Hospital, Hull, UK
[5]Institute of Cancer Research, London, UK
[6]The Department of Haemato-oncology, Royal Marsden Hospital NHS Trust, London, UK
[7]Haematology Malignancy Diagnostic Service (HMDS), St James's University Hospital, Leeds, UK
[8]Department of Haematology, University College London Hospitals NHS Foundation Trust, London, UK
[9]King's College Hospital, London, UK
[10]Brighton and Sussex Medical School, Brighton, UK
[11]Department of Haematology, Leeds Teaching Hospitals NHS Trust, Leeds, UK
[12]Department of Haematology and Stem Cell Transplantation, University Hospitals Birmingham NHS Foundation Trust, Birmingham, UK
[13]Academic Unit of Health Economics, University of Leeds, Leeds Institute of Health Sciences, Leeds, UK
[14]Southampton General Hospital, Southampton, UK
[15]Institute of Immunology and Immunotherapy, Department of Haematology, University of Birmingham, Birmingham, UK
[16]Department of Haematology, Newcastle University, Newcastle upon Tyne, UK
[17]Leeds Cancer Centre, St James's University Hospital, Leeds, UK

**Contributors** The FiTNEss trial was conceived by GC and GJ and designed by them in collaboration with CPawlyn, DAC, AH, JB, SB, MK, RO, MJ, BK, and MD. All authors (ABC, K-LR, CPawlyn, DAC, AH, JB, SB, MK, RdT, NR, KB, JJ, CParrish, HG, DM, BD, CO, RH, PB, RO, MJ, BK, MD, GJ and GC) inputted into the development of the protocol and patient information sheet. The first draft of the manuscript was written by ABC and K-LR. All authors (ABC, K-LR, CPawlyn, DAC, AH, JB, SB, MK, RdT, NR, KB, JJ, CParrish, HG, DM, BD, CO, RH, PB, RO, MJ, BK, MD, GJ and GC) reviewed and approved the final manuscript.

**Funding** This trial is funded by Cancer Research UK (C37712/A21282) with further unrestricted funding from Millennium: The Takeda Oncology Company and Celgene: A Bristol Myers Squibb Company. This work was also supported by Core Clinical Trials Unit Infrastructure from Cancer Research UK (C7852/A25447). The trial Sponsor is responsible for the overall conduct of the trial as defined by Directive 2001/20/EC is University of Leeds, UoL/LTHT Joint Sponsor QA office (CTIMPs); Research & Innovation Centre/Faculty of Medicine & Health; Leeds Teaching Hospitals NHS Trust/University of Leeds; St James University Hospital; Leeds LS9 7TF.

**Disclaimer** The funders had no role in the design, collection, analysis or collection of data; in writing the manuscript; or in the decision to submit the manuscript for publication.

**Competing interests** ABC, K-LR, DAC, AH, CO, RH and PB report grants and non-financial support from BMS/Celgene, grants and non-financial support from Merck Sharpe & Dohme, grants and non-financial support from Amgen, grants and non-financial support from Takeda, during the conduct of the trial. DAC also reports travel support from Celgene Corporation. CPawlyn reports receiving honoraria and/or travel support from Amgen, BMS/Celgene, Janssen, Sanofi and Takeda. MK consultancy: AbbVie, Amgen, BMS/Celgene, GSK, Janssen, Karyopharm, Seattle Genetics, Takeda; honoraria: BMS/Celgene, Janssen, Takeda; Research funding: BMS/Celgene; Travel/educational support: BMS/Celgene, Janssen, Takeda. SB reports receiving research funding from Takeda. KB—Advisory Boards Janssen: BMS/Celgene, Takeda, Novartis. Speaker Honoraria: Janssen, BMS/Celgene, Sanofi, Takeda. Support to attend educational meetings: Janssen, BMS/Celgene, Takeda, GSK. GJ reports research funding from Takeda, Onyx, MSD & BMS/Celgene with consultancy from Janssen, Takeda, Sanofi, Oncopeptides, Karyopharm, Pfizer, Roche & BMS/Celgene. GC reports research funding from Janssen, Takeda, Amgen & BMS/Celgene with consultancy from Janssen, Takeda, Sanofi, Oncopeptides, Karyopharm, Pfizer, Roche & BMS/Celgene. MD reports Stock held in Abingdon Health. JB, RdT, NR, JJ, CParrish, HG, DM, BD, RO, MJ and BK have no declared competing interests.

**Patient and public involvement** Patients and/or the public were involved in the design, or conduct, or reporting, or dissemination plans of this research. Refer to the Methods section for further details.

**Patient consent for publication** Not applicable.

**Provenance and peer review** Not commissioned; externally peer reviewed.

of the author(s) and are not endorsed by BMJ. BMJ disclaims all liability and responsibility arising from any reliance placed on the content. Where the content includes any translated material, BMJ does not warrant the accuracy and reliability of the translations (including but not limited to local regulations, clinical guidelines, terminology, drug names and drug dosages), and is not responsible for any error and/or omissions arising from translation and adaptation or otherwise.

**ORCID iDs**
Amy Beth Coulson http://orcid.org/0000-0002-1810-409X
Kara-Louise Royle http://orcid.org/0000-0003-0225-1199

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
