## [Reviewer comments · BMJ Open]

ARTICLE DETAILS

TITLE (PROVISIONAL)	Frailty-adjusted therapy in Transplant Non-Eligible patients with newly diagnosed Multiple Myeloma (FiTNEss (UK-MRA Myeloma XIV Trial)): A study protocol for a randomised phase III trial
AUTHORS	Coulson, Amy; Royle, Kara-Louise; Pawlyn, Charlotte; Cairns, David; Hockaday, Anna; Bird, Jennifer; Bowcock, Stella; Kaiser, Martin; de Tute, Ruth; Rabin, Neil; Boyd, Kevin; Jones, John; Parrish, Christopher; Gardner, Hayley; Meads, David; Dawkins, Bryony; Olivier, Catherine; Henderson, Rowena; Best, Phillip; Owen, Roger; Jenner, Matthew; Kishore, Bhuvan; Drayson, Mark; Jackson, Graham; Cook, Gordon

VERSION 1 – REVIEW

REVIEWER	Lad, Deepesh Postgraduate Institute of Medical Education and Research
REVIEW RETURNED	30-Sep-2021

GENERAL COMMENTS	Excellent study protocol to address the unmet need of the frail myeloma patient population. This study is the first of its kind and will be practice-changing. I have few minor comments. 1. Since the criteria for transplant ineligibility varies from center to center and the study mentions that the participants will be TNE as assessed by their treating clinician, it will be helpful if this is defined for uniformity in the study.2. In the adaptive dosing arm, since the lenalidomide dose will be decided by frailty as well as for renal impairment, please clarify how will be the dosing for patients with fit patients with renal impairment. A frail patient with renal impairment may still end up getting the maximum recommended dose as per his renal function.3. participants in the adaptive arm of R1 will have their dose adjusted according to changes in the frailty category at the start of cycles 3, 5, and 7. Doses can also be escalated for suboptimal responders under certain criteria. How will such dynamic frailty and dose changes be considered in the statistical analysis?4. Primary outcome: Participants will be defined to have experienced an event if they die, progress, or are withdrawn from treatment (by the treating clinician) or withdraw consent for treatment, within 60 days of R1. Why is the primary outcome assessed within 60 days of R1? Since frailty and the dosing is dynamic at time points after 2,4,6 and 12 cycles, the primary outcome would be most important after completion of induction (12 cycles). That would be the ideal time point even for the real world, where this vulnerable population would withdraw from treatments due to toxicity or progression.
---

	5. It will be useful to stratify randomization using the R-ISS rather than Beta-2 microglobulin concentration, Hemoglobin, creatinine, calcium, platelets.
--	--

REVIEWER	Bonello, Francesca University of Torino
REVIEW RETURNED	15-Oct-2021

GENERAL COMMENTS	This trial represents one of the largest randomized trial exploring a frailty-adapted treatment approach, and its results are eagerly awaited. Just two minor points to be addressed: 1. Line 42-42, page 9: does this mean that if a patients changes frailty category for age only (eg 80 years old unfit turns 81 and is categorized as frail) the treatment dose is reduced, even in the absence of toxicity? Could a dynamic change in frailty status within the induction phase represent a bias while interpreting study results? 2. Line 13, page 11: how and at which timepoints is MRD assessed. Is sustained MRD negativity evaluated? Is the conversion rate from MRD pos to ne during maintenance analysed?
--

VERSION 1 – AUTHOR RESPONSE

Reviewer 1:

Excellent study protocol to address the unmet need of the frail myeloma patient population. This study is the first of its kind and will be practice-changing. I have few minor comments.

Minor Comments

1) Since the criteria for transplant ineligibility varies from center to center and the study mentions that the participants will be TNE as assessed by their treating clinician, it will be helpful if this is defined for uniformity in the study.

In the FITNEss trial, the decision as to whether the participant is transplant non-eligible (TNE) was left at the discretion of the Principle Investigator at each site, based on age and co-morbidities. Most centres in the UK do not assess transplant eligibility against formalised criteria and the decision is made on a case by case basis. Although transplant scoring systems are published these are not currently standardised and are not in widespread use in the UK or required for enrolment in the study. As a general rule in the UK patients aged less than 65 years old are considered transplant eligible unless they have significant co-morbidities that would preclude this approach. Patients aged over 75 years old tend to be considered transplant ineligible and patients between 65-75 years old will have the decision based on patient and clinician preference, co-morbidities and patient fitness. Unlike other late phase trials in transplant non-eligible patients, the use of IMWG frailty scoring and MRP helps control for this meaning the population in FITNEss will be more homogenous than previously reported in other phase III TNE myeloma trials.

2) In the adaptive dosing arm, since the lenalidomide dose will be decided by frailty as well as for renal impairment, please clarify how will be the dosing for patients with fit patients with renal

impairment. A frail patient with renal impairment may still end up getting the maximum recommended dose as per his renal function.

As per standard of care, doses of lenalidomide and ixazomib should be adjusted at baseline according to liver and renal failure. Regardless of which dosing arm has been assigned the doses indicated to be taken due to renal impairment should not be exceeded on cycle 1 day 1. For a fit patient assigned to the adaptive dosing arm, lenalidomide should be given as follows:

- Moderate renal impairment – 10mg once daily
- Severe renal impairment – 15mg every other day
- End of stage renal disease – 5mg once daily (on dialysis days should be administered following dialysis)

Doses of lenalidomide reduced (based on renal function) can only be escalated, if tolerated, to the maximum dose indicated by the participants assigned dosing arm/ strategy after cycle 1 as per the lenalidomide SPC.

A Frail patient assigned to the adaptive dosing arm, where lenalidomide is given 10mg daily, may end up getting the maximum recommended dose if they have with moderate renal impairment.

We have clarified in the protocol paper the dosing of lenalidomide due to renal and liver function. Further information has been added to the footnote under the 'Interventions and Dosing section' of the protocol paper (Page 9). In addition we have also included the dosing schedule for renal and liver function in the supplementary material.

3) Participants in the adaptive arm of R1 will have their dose adjusted according to changes in the frailty category at the start of cycles 3, 5, and 7. Doses can also be escalated for suboptimal responders under certain criteria. How will such dynamic frailty and dose changes be considered in the statistical analysis?

The objective of this study is to investigate the adaptive dosing strategy as a whole, encompassing any changes in frailty and allowances for sub-optimal responders. Therefore at present no additional sensitivity analysis is planned for the primary or secondary endpoints. Instead, changes in frailty will be summarised descriptively and the total amount of therapy participants receive will be investigated through the secondary endpoint "treatment compliance and total amount of therapy delivered".

4) Primary outcome: Participants will be defined to have experienced an event if they die, progress, or are withdrawn from treatment (by the treating clinician) or withdraw consent for treatment, within 60 days of R1. Why is the primary outcome assessed within 60 days of R1? Since frailty and the dosing is dynamic at time points after 2,4,6 and 12 cycles, the primary outcome would be most important after completion of induction (12 cycles). That would be the ideal time point even for the real world, where this vulnerable population would withdraw from treatments due to toxicity or progression.

In the design of our study we considered the peri-randomisation phase to be most critical in keeping patients on treatment and chose this early endpoint to reflect this. An event up to 60 days was chosen as this was shown to be a critical period in the care of myeloma patients in a comprehensive analysis of trial participants published by our trials collaborative (<https://pubmed.ncbi.nlm.nih.gov/16275935/>). This period has also shown evidence of being high-risk for TNE patients in MRC Myeloma IX and NCR1 Myeloma XI, with no improvement in clinical outcomes between these studies in the early stages of follow-up (Figure 1B). We will also consider the frequency of these events after this timepoint and report it alongside other secondary endpoints which have a longer time horizon.

5) It will be useful to stratify randomization using the R-ISS rather than Beta-2 microglobulin concentration, Hemoglobin, creatinine, calcium, platelets.

We chose these stratification factors to be consistent with our earlier trials (MRC Myeloma IX and NCR1 Myeloma XI) to allow easy cross comparison of relevant subgroups. In addition the use of R-ISS as a stratification factor can prove challenging, due to it being difficult and time-consuming to collect. In order to calculate R-ISS, certain biological features of the patients multiple myeloma need to be known (i.e. standard risk and high risk disease defined according to <https://www.ncbi.nlm.nih.gov/pmc/articles/PMC4846284/pdf/zlj2863.pdf>). One aspect of this definition includes the presence of pre-defined chromosomal abnormalities determined by FISH (Fluorescent in situ hybridization). Local centres on the study may not routinely test for all of these abnormalities; therefore testing would need to be conducted centrally in real time, which is inefficient as it increases trial costs as well as delays to treatment. R-ISS will instead be derived as part of the statistical analysis, and reported alongside the trial results.

Reviewer 2:

This trial represents one of the largest randomized trial exploring a frailty-adapted treatment approach, and its results are eagerly awaited.

Minor Comments

1) Line 42-42, page 9: does this mean that if a patients changes frailty category for age only (eg 80 years old unfit turns 81 and is categorized as frail) the treatment dose is reduced, even in the absence of toxicity? Could a dynamic change in frailty status within the induction phase represent a bias while interpreting study results?

We apologise for the lack clarity in this section. When calculating the Myeloma Frailty Index of the participant, the participant's age will be consistently used as the age at the time of (Main) Trial Registration. This means that a patient's frailty category will never change from age alone. We have added a footnote to this section to clarify how age is used in the frailty index.

2) Line 13, page 11: how and at which timepoints is MRD assessed. Is sustained MRD negativity evaluated? Is the conversion rate from MRD post to ne during maintenance analysed?

As detailed within the central sample analysis section of the supplementary material MRD will be assessed in bone marrow aspirates using validated flow cytometry assay (sensitivity $\leq 0.004\%$) performed at a single central laboratory (HMDS, Leeds Teaching Hospitals Trust). A minimum of 500,000 cells will be evaluated with six- or eight-colour antibody combinations including CD138, CD38, CD45, CD19, CD56, CD27, CD81 and CD117.

Sustained MRD-negativity, as defined by the IMWG

(<https://www.sciencedirect.com/science/article/pii/S1470204516302066>), will not be assessed as it requires the participant to be imaging MRD-negative at the same timepoints that they are flow MRD-negative and imaging is only mandated at baseline and to investigate new systems not at regular intervals for all participants.

Flow MRD will be assessed at the following time points: Baseline (pre-randomisation 1), after 6 cycles of induction therapy, at the end of 12 cycles of induction treatment (or at the end of the final cycle of IRD treatment if sooner), at 12 months post maintenance randomisation 2. Flow MRD will also be assessed at any point a first occurrence of CR, or SCR is suspected.

During final analysis, we will determine the conversion rate of patients who are MRD Positive at the end of induction to MRD Negative at 12 months post Randomisation 2. The analysis to this is specified within our Statistical Analysis Plan (SAP).

This additional detail regarding sustained MRD testing, time points and analysis has been added to the supplementary material (Central Lab Analysis; HMDS, Page 21). A footnote has also been added to page 11 in the protocol paper.

VERSION 2 – REVIEW

REVIEWER	Lad, Deepesh Postgraduate Institute of Medical Education and Research
REVIEW RETURNED	26-Jan-2022
GENERAL COMMENTS	All queries have been answered.